# Unique characteristics of new complete blood count parameters, the Immature Platelet Fraction and the Immature Platelet Fraction Count, in dengue patients

Ikkoh Yasuda[1,2,3], Nobuo Saito[1,2,4]*, Motoi Suzuki[2,5], Dorcas Valencia Umipig[6], Rontgene M. Solante[6], Ferdinand De Guzman[6], Ana Ria Sayo[6], Michio Yasunami[2,7], Nobuo Koizumi[8], Emi Kitashoji[2], Kentaro Sakashita[1,9], Chris Fook Sheng Ng[1], Chris Smith[1,10], Koya Ariyoshi[1,2]

1 School of Tropical Medicine and Global Health, Nagasaki University, Nagasaki, Japan, 2 Department of Clinical Medicine, Institute of Tropical Medicine, Nagasaki University, Nagasaki, Japan, 3 Department of General Internal Medicine and Clinical Infectious Diseases, Fukushima Medical University, Fukushima, Japan, 4 Department of Microbiology, Oita University Faculty of Medicine, Oita, Japan, 5 Infectious Disease Surveillance Center, National Institute of Infectious Diseases, Tokyo, Japan, 6 San Lazaro Hospital, Manila, The Philippines, 7 Saga-Ken Medical Centre Koseikan, Saga, Japan, 8 Department of Bacteriology I, National Institute of Infectious Diseases, Toyama, Shinjuku-ku, Tokyo, Japan, 9 Department of Basic Mycobacteriology, Graduate School of Biomedical Science, Nagasaki University, Nagasaki, Japan, 10 Department of Clinical Research, London School of Hygiene and Tropical Medicine, London, United Kingdom

* nobuosaito@oita-u.ac.jp

**Data Availability Statement:** Our minimal data set is available at the following URL: https://doi.org/10.6084/m9.figshare.16810771.v1.

## Abstract

The advanced platelet parameters Immature Platelet Fraction and Immature Platelet Fraction Count have been implemented in clinical practice as measures of thrombopoietic activity, mainly in hematologic disorders that cause thrombocytopenia. The purpose of this observational study was to examine thrombopoiesis as reflected by these 2 new CBC parameters in patients infected with dengue. The study was conducted in infectious disease referral hospital in Metro Manila, the Philippines. We enrolled hospitalized patients at admission who were diagnosed with acute dengue or community acquired bacterial infection (CABI). Immature Platelet Fraction (IPF) and Immature Platelet Fraction Count were evaluated at admission and during hospitalization. A total of 606 patients were enrolled from May 1, 2017 to June 1, 2018. The participants consisted of 152 patients with dengue infection, 180 confirmed CABI, and 274 suspected CABI patients. At admission, the percent IPF (IPF %) of the patients with dengue was significantly higher than that of the confirmed CABI patients (median 3.7% versus 1.9%; p <0.001). In a time course evaluation, there was no significant difference of IPF% between the patients with dengue infection and the confirmed CABI patients in the febrile phase (median 1.9% versus 2.4%; p = 0.488), however, the IPF % of the patients with dengue infection increased to be significantly higher than that of the confirmed CABI patients in the critical phase (median 5.2% versus 2.2%; p <0.001). Our study elucidated the unique characteristics and time-course trends of IPF percent and number (IPF#) in the patients with dengue infection. IPF% and IPF# are potentially valuable

**Funding:** This work was mostly supported by the Ministry of Education, Culture, Sports, Science, and Technology (MEXT), Government of Japan. Department of Clinical Medicine, Institute of Tropical Medicine, Nagasaki University has received research fund from Sysmex Corporation for the study: "An observational study of community acquired-bacteremia in San Lazaro Hospital, Manila, the Philippines" and the costs of testing for this study were partially covered by the research fund. The funders had no role in study design, data collection and analysis, decision to publish, or preparation of the manuscript. URL of each funder website: https://www.mext.go.jp/en/ https://www.sysmex.co.jp/en/index.html.

**Competing interests:** The authors have no conflicts of interest associated with Sysmex Corporation relating to the employment, consultancy, patents, products in development, and marketed products. This does not alter our adherence to PLoS ONE policies on sharing data and materials.

parameters in dengue and further investigation is required for the optimal use in clinical practice.

## Introduction

Dengue is a mosquito-borne viral infection and is one of the most important viral diseases in tropical areas [1, 2]. A total of 390 million dengue infections are estimated to occur per year, and 3.97 billion people are estimated to be at risk of dengue infection worldwide [3, 4]. Thrombocytopenia induced by infection with dengue is typical around the time of defervescence and is recognized as a potential indicator of clinical worsening according to the 2009 WHO guidelines [1, 2]. However, the underlying pathophysiological mechanisms involved in dengue-induced thrombocytopenia remain controversial [5]. Recently, the Immature Platelet Fraction (IPF%) and Immature Platelet Fraction Count (IPF#) have been recognized as measures of thrombopoietic activity [6, 7]. Conceptually, IPF% and IPF# correspond respectively to the percentage and absolute number of immature platelets in peripheral blood. Various studies have evaluated the utility of these parameters in the evaluation of patients with haematological conditions such as idiopathic thrombocytopenic purpura (ITP), thrombotic thromboctopenic purpura (TTP), aplastic anaemia and chemotherapeutic related thrombocytopenia [6, 8]. It is assumed that an increased IPF% indicates a consumptive or destructive thrombocytopenic status; whereas a normal or decreased IPF% suggests decreased platelet production in bone marrow [9–11]. IPF# is considered to reflect real-time platelet production [7]. Although IPF% and IPF# have been implemented in wider clinical settings, the benefit of evaluating these parameters in patients with dengue infection has not been determined. Therefore, this study aimed to investigate the thrombopoietic activity in patients with dengue infection by quantifying IPF% and IPF# and to elucidate their characteristics by comparing with these parameters in patients with community acquired bacterial infection (CABI). CABI was chosen as the control group because the main topic of this study was the utility of IPF among febrile patients with thrombocytopenia. CABI-associated thrombocytopenia has clinically significant differences from dengue; specifically, the need for antibiotic treatment and a poorer prognosis.

## Materials and methods

This analysis was implemented under an existing study: "An observational study of community acquired-bacteremia in San Lazaro Hospital, Manila, The Philippines". San Lazaro Hospital is a national tertiary referral and training hospital for infectious diseases and has a 500-bed capacity. This substudy comprised patients who were enrolled in the main study from May 1, 2017 to June 1, 2018 and satisfied eligibility criteria as follows: (i) admitted to San Lazaro Hospital, (ii) aged $\geq$ 1 year at admission, and (iii) having acute onset of fever ($\leq$ 21 days) at admission. The participants were subsequently categorized into the three groups—confirmed dengue infection, confirmed CABI, or suspected CABI–following the disease definitions described below. Because limited diagnostic techniques were available at the study site and few varieties of bacterial infectious diseases could be definitively diagnosed, the confirmed CABI group was potentially weighted toward the diseases that were definitively diagnosable at the study site. Therefore, the suspected CABI was included as a category to avoid selection bias. Subjects were excluded if suspected of active tuberculosis or diagnosed with human immunodeficiency virus (HIV) infection. Participants were also excluded if unable to provide blood for a complete blood count (CBC) test at admission or who received blood transfusion during hospitalization.

## Disease definition

Dengue diagnosis was established when the nonstructural protein 1 (NS1) antigen was positive and/or dengue reverse transcriptase polymerase chain reaction (PCR) was positive without a positive blood culture result. Patients were also diagnosed with dengue if positive for dengue IgM without laboratory positive results of other diseases or positive blood culture results. Diagnostic criteria of each CABI are described in S1 Appendix. The CABI that fulfilled the diagnostic criteria was categorize as confirmed CABI. Suspected CABI was a clinical diagnosis after excluding dengue infection but unable to reach a confirmed CABI diagnosis described in S1 Appendix.

Severe thrombocytopenia was defined as platelets $<50\times10^3/\mu l$ at admission or a platelet nadir $<50\times10^3/\mu l$ during hospitalization in a time course evaluation. Non-severe thrombocytopenia was defined to include all patients with platelets $\geq50\times10^3/\mu l$ at admission or a platelet nadir $\geq50\times10^3/\mu l$ during hospitalization in a time course evaluation.

Severe dengue was clinically defined as having at least one of the following conditions at admission: systolic blood pressure $\leq90$ mmHg, desaturation requiring oxygenation, aspartate aminotransferase $\geq 1000$ IU/L and/or alanine transaminase $\geq1000$ IU/L, or impaired consciousness. Anemia was defined by hemoglobin levels at admission according to WHO guidelines [12].

NS1 antigen and dengue IgM were tested using a SD BIOLINE Dengue Duo™ kit (Standard Diagnostics, Korea). Laboratory procedure for diagnoses of CABIs are described in S2 Appendix. The details of PCR and ELISA method used in this study were as published [13–17].

## Laboratory procedure for IPF% and IPF#

Peripheral blood samples were drawn into tubes containing ethylene diamine tetra-acetic acid (EDTA). In addition to regular CBC parameters, IPF% and IPF# were evaluated using an automated hematology analyzer (Sysmex XN-1000™, Sysmex, Kobe, Japan). The analyzer detects immature platelets, which are larger in size and contain more RNA than mature platelets, by staining intracellular RNA with oxazine fluorescent dye. IPF% is expressed as a percentage representing the ratio of the absolute number of immature platelets to the total number of platelets. IPF# represents the absolute number of immature platelets per unit volume.

## Data collection and statistical analysis

Age, demographic data and past medical history were documented at admission. The results of the admission and subsequent CBC results of the patients were recorded in an electronic data base. To evaluate the time course trend of platelet parameters, the representative values of each parameter by day of illness were calculated using the results of participants who underwent blood tests on a specific day of illness. The timing of blood collection varied between participants and the sequential day-by-day test results were not available for everyone. We defined the days of illness as the duration from the day of onset defined as day 0 of illness to the day of interest. The clinicians or the researchers were not blinded to the laboratory results.

Baseline characteristics were compared using Fisher's exact test for categorical variables or the Mann Whitney test for continuous variables. Comparison of the results of dengue, confirmed CABI, and suspected CABI patients were performed using the Kruskal-Wallis test followed by the Dunn's post hoc test with Holm adjustment. Pairwise comparisons were performed using the Mann Whitney test. P values of less than 0.05 were considered statistically significant. All analyses were performed using Stata version 14.2 (Stata Corp., College Station, TX, USA).

### Ethical issues

This work was conducted as a sub-study of a main study: "An observational study of community acquired bacteremia in San Lazaro Hospital, Manila, The Philippines". Ethical approval for the main study was obtained from the Research Ethical and Review Unit of San Lazaro Hospital, the Philippines (number: SLH-RERU-2015-005-E) and the Institutional Review Board of the Institute of Tropical Medicine, Nagasaki University, Japan (number: 150226136–4). Written informed consent was obtained from guardians or caregivers for patients aged under 18 years of age, illiterate or unconscious at presentation. The requirements of the institutional review boards during the study period included obtaining written informed consent; however, they did not include obtaining assent of the patients for whom guardians or caregivers provided written consent. For all others, written consent was obtained from the participants. The institutional review boards approved the consent procedures.

## Results

### 1. Basic characteristics

The characteristics of participants at admission are summarized in Table 1. A total of 606 patients were eligible after excluding 19 subjects who satisfied the inclusion criteria but received blood transfusion during treatment and one participant whose CBC test result was not available at admission. The participants consisted of 152 patients suffering with dengue, 180 confirmed CABI and 274 suspected CABI patients. The confirmed diagnosis in the CABI patients included leptospirosis (59 patients), X-ray confirmed pneumonia (37 patients), bacteremia (12 patients), diphtheria (25 patients), meningococcal disease (14 patients), and skin infection (33 patients). The clinical diagnoses in the suspected CABI patients consisted of pneumonia (72 patients), enteric fever (26 patients), urinary tract infection (23 patients), leptospirosis (10 patients), central nervous system infection (8 patients), abdominal infection (4 patients), meningococcal disease (1 patient), septic rash (1 patient), and undiagnosable infectious disease (129 patients). By definition, all suspected CABIs were diagnosed clinically and no suspected CABI patient fulfilled the diagnostic criteria of confirmed CABIs. The median age was 19.0 years (interquartile range (IQR): 13.0, 25.0) in the dengue group, 20.5 years (IQR: 9.0, 33.5) in the confirmed CABI group and 19.0 years (IQR: 10.0, 32.0) in the suspected CABI group, and there were no significant differences between the dengue and each CABI group.

### 2. Comparison of platelet, IPF%, and IPF# at admission between the dengue, confirmed CABI, and suspected CABI groups

Fig 1 shows the comparison of platelet, IPF% and IPF# at admission between the dengue, confirmed CABI and suspected CABI groups. The median day of illness at admission was 5 in all groups and there was no significant difference among groups. (A) shows a comparison including all participants. Platelet counts of the dengue group were significantly lower than those of the confirmed CABI (median $88.0\times10^3$/μL versus $225.5\times10^3$/μL; p <0.001) and suspected CABI groups (median $88.0\times10^3$/μL versus $221.0\times10^3$/μL; p <0.001). The IPF% of the dengue group was significantly higher than those of the confirmed CABI (median 3.7% versus 1.9%; p <0.001) and suspected CABI groups (median 3.7% versus 1.9%; p <0.001). Although there was no significant difference of IPF# between the dengue group and the confirmed CABI group (median $3.1\times10^3$/μL versus $3.9\times10^3$/μL; p = 0.057), the IPF# of the dengue group was significantly lower than that of the suspected CABI group (median $3.1\times10^3$/μL versus $3.6\times10^3$/μL; p = 0.005). The platelet parameters of each confirmed CABI are shown separately in S1 Fig. (B) shows a comparison including only the subgroups with severe thrombocytopenia

**Table 1. Basic characteristics of participants at admission.**

|  |  | Dengue (n = 152) | Confirmed CABI (n = 180) | p* | Suspected CABI (n = 274) | p* |
|---|---|---|---|---|---|---|
|  |  | n (%) or median (IQR) | n (%) or median (IQR) |  | n (%) or median (IQR) |  |
| Age |  | 19.0 (13.0, 25.0) | 20.5 (9.0, 33.5) | 0.628 | 19.0 (10.0, 32.0) | 0.985 |
| Male |  | 100 (65.8) | 133 (73.9) | 0.118 | 175 (63.9) | 0.751 |
| Mortality |  | 1 (0.7) | 22 (12.2) | <0.001 | 9 (3.3) | 0.104 |
| Co-morbid conditions |  | 17 (11.2) | 42 (23.3) | 0.004 | 78 (28.5) | <0.001 |
| Days of illness at admission |  |  |  |  |  |  |
| 0–3 |  | 40 (26.3) | 42 (23.3) | 0.027 | 75 (27.4) | <0.001 |
| 4,5 |  | 71 (46.7) | 63 (35.0) |  | 78 (28.5) |  |
| ≥6 |  | 38 (25.0) | 72 (40.0) |  | 114 (41.6) |  |
| unknown onset |  | 3 (2.0) | 3 (1.7) |  | 7 (2.6) |  |
| Patients with anemia at admission |  | 23 (15.1) | 93 (51.7) | <0.001 | 94 (34.3) | <0.001 |
| **Routine hematology at admission** |  |  |  |  |  |  |
| Platelets | ($10^3$/μL) | 88.0 (48.5, 161.5) | 225.5 (100.0, 342.5) | <0.001 | 221.0 (141.0, 316.0) | <0.001 |
| IPF% | (%) | 3.7 (1.6, 7.2) | 1.9 (0.9, 3.5) | <0.001 | 1.9 (1.0, 3.7) | <0.001 |
| IPF# | ($10^3$/μL) | 3.1 (1.9, 4.9) | 3.9 (1.5, 7.0) | 0.084 | 3.6 (2.4, 5.8) | 0.002 |
| WBC | ($10^3$/μL) | 4.2 (2.6, 6.5) | 12.4 (8.3, 16.4) | <0.001 | 8.3 (5.7, 12.6) | <0.001 |
| Neutrophils | ($10^3$/μL) | 1.8 (1.1, 3.3) | 9.9 (5.9, 13.8) | <0.001 | 5.5 (3.3, 9.4) | <0.001 |
|  | (%) | 55.2 (35.5, 70.8) | 79.0 (67.0, 87.5) | <0.001 | 69.7 (54.6, 79.4) | <0.001 |
| Lymphocytes | ($10^3$/μL) | 1.1 (0.6, 2.4) | 1.5 (0.7, 2.1) | 0.543 | 1.6 (1.1, 2.4) | <0.001 |
|  | (%) | 33.3 (20.0, 51.8) | 12.5 (5.5, 21.5) | <0.001 | 20.2 (12.1, 32.8) | <0.001 |
| Monocytes | ($10^3$/μL) | 0.4 (0.2, 0.6) | 0.8 (0.5, 1.1) | <0.001 | 0.7 (0.5, 1.1) | <0.001 |
|  | (%) | 9.4 (6.6, 13.2) | 6.4 (4.5, 8.8) | <0.001 | 8.2 (5.9, 10.6) | 0.002 |
| Eosinophils | ($10^3$/μL) | 0.01 (0.00, 0.05) | 0.04 (0.01, 0.19) | <0.001 | 0.03 (0.00, 0.14) | <0.001 |
|  | (%) | 0.4 (0.0, 1.0) | 0.3 (0.0, 1.5) | 0.136 | 0.3 (0.0, 1.9) | 0.085 |
| RBC | ($10^6$/μL) | 5.1 (4.7, 5.6) | 4.5 (4.1, 4.9) | <0.001 | 4.7 (4.4, 5.2) | <0.001 |
| Hemoglobin | (g/dL) | 138.0 (127.0, 152.5) | 122.0 (108.5, 132.0) | <0.001 | 126.0 (115.0, 140.0) | <0.001 |
| Hematocrit | (%) | 40.9 (38.7, 45.0) | 36.9 (33.0, 40.0) | <0.001 | 38.5 (34.6, 42.2) | <0.001 |
| AST | (IU/L) | 97.0 (57.0, 168.0) | 32.0 (22.0, 59.0) | <0.001 | 48.5 (28.0, 110.5) | <0.001 |
| ALT | (IU/L) | 58.0 (32.0, 118.0) | 29.5 (20.0, 57.0) | <0.001 | 44.0 (22.0, 82.0) | 0.029 |
| BUN | (mg/dl) | 9.8 (7.1, 12.6) | 15.1 (10.2, 39.0) | <0.001 | 9.4 (6.9, 13.8) | 0.823 |
| Creatinine | (mg/dl) | 0.8 (0.6, 0.9) | 1.1 (0.6, 2.3) | <0.001 | 0.8 (0.6, 1.1) | 0.366 |
| CRP | (mg/L) | 0.6 (0.1, 1.4) | 11.5 (5.3, 16.0) | <0.001 | 2.8 (0.9, 9.4) | <0.001 |
| PCT | (ng/mL) | 0.6 (0.3, 1.2) | 2.2 (0.3, 10.0) | <0.001 | 0.4 (0.1, 1.8) | 0.052 |

\* Between dengue and each CABI group using Fisher's exact test for categorical variables or Mann Whitney test for continuous variables.

Missing number of participants in the dengue, confirmed CABI and suspected CABI group for AST = 89, 107, 170; ALT = 78, 84, 156; BUN = 83, 81, 153; Cre = 70, 71, 151; PCT = 6, 6, 17. CABI: community acquired bacterial infection, IQR: interquartile range, IPF%: Immature Platelet Fraction, IPF#: Immature Platelet Fraction Count. WBC: white blood cell, RBC: red blood cell, AST: aspartate aminotransferase, ALT: alanine aminotransferase, BUN: blood urea nitrogen, CRP: C-reactive protein, PCT: procalcitonin.

<50×$10^3$/μL at admission. The median day of illness at admission was 5 for the three severe thrombocytopenia groups and there was no significant difference among groups. There was no significant difference of platelet counts among the three groups. Although the IPF% of the dengue group remained significantly higher than that of the confirmed CABI group (median 9.5% versus 3.4%; p = 0.005), there was no significant difference of IPF% between the dengue and the suspected CABI group (median 9.5% versus 6.9%; p = 0.247). The IPF# of the dengue group was significantly higher than that of the confirmed CABI group (median 2.5×$10^3$/μL

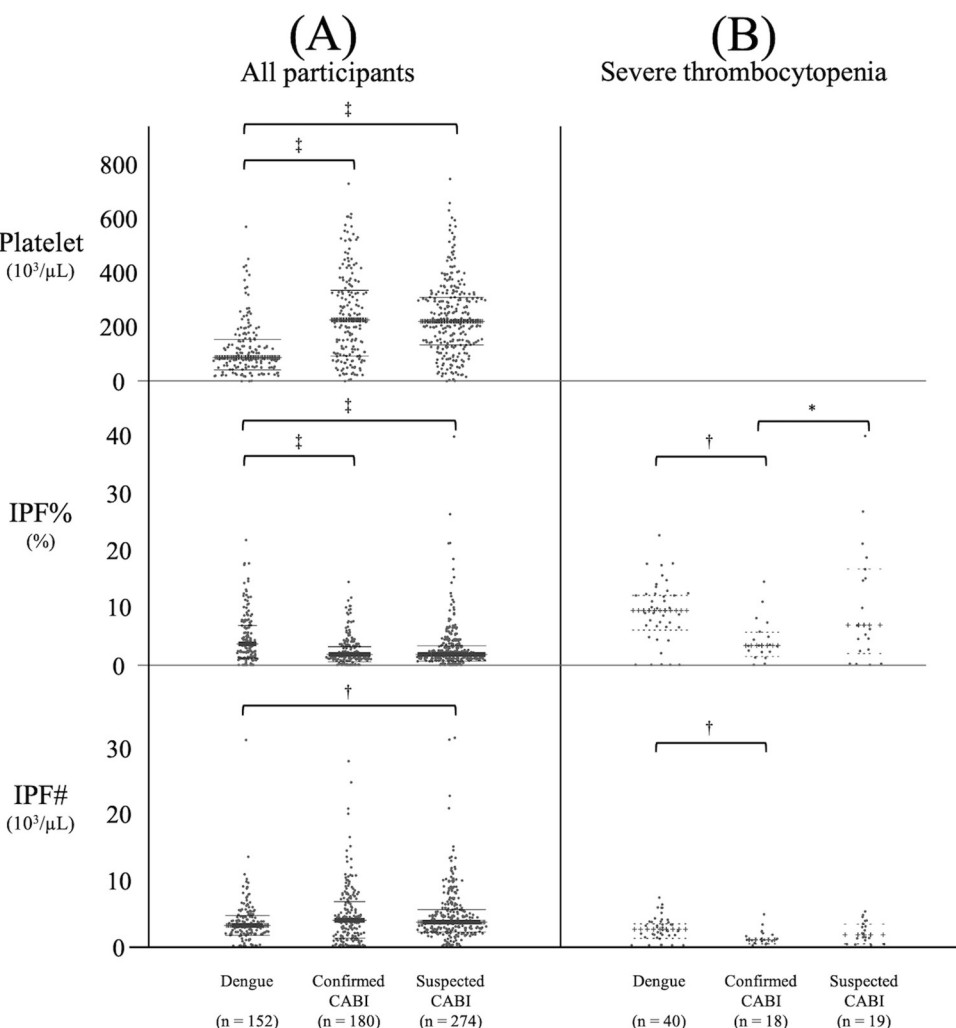

**Fig 1. Comparison of platelet, IPF% and IPF# among dengue, confirmed CABI, and suspected CABI groups at admission.** Horizontal lines show the median and interquartile ranges. (A) shows a comparison including all participants and (B) shows a comparison including only the subgroups with severe thrombocytopenia <50×10³/μL at admission. Comparison of dengue, confirmed CABI and suspected CABI groups were performed using Kruskal-Wallis test followed by the Dunn's post hoc test with Holm adjustment. *: P<0.05, †: P<0.01, ‡: P<0.001. CABI: community acquired bacterial infection, IPF%: Immature Platelet Fraction, IPF#: Immature Platelet Fraction Count.

versus $0.8×10^3$/μL; P = 0.007) but there was no significant difference of the IPF# between the dengue and the suspected CABI group (median $2.5×10^3$/μL versus $1.6×10^3$/μL; p = 0.123).

## 3. Time course of platelet parameters by days of illness

Fig 2 shows the time course trends of platelet, IPF% and IPF# in the dengue, confirmed CABI and suspected CABI groups from the first to tenth days of illness. The platelet counts of the dengue group decreased from day 1, with a minimum observed on the sixth day of illness before recovering to a normal level approximately on the tenth day of illness. The platelet count of the two CABI groups shared a similar pattern, with a slightly increasing trend for the whole duration. The IPF% of the dengue group increased from the start and peaked on the sixth and seventh days, mirroring the pattern of the platelet count. The IPF% of the two CABI groups remained level during the whole period. The IPF# of all three groups showed a gradual

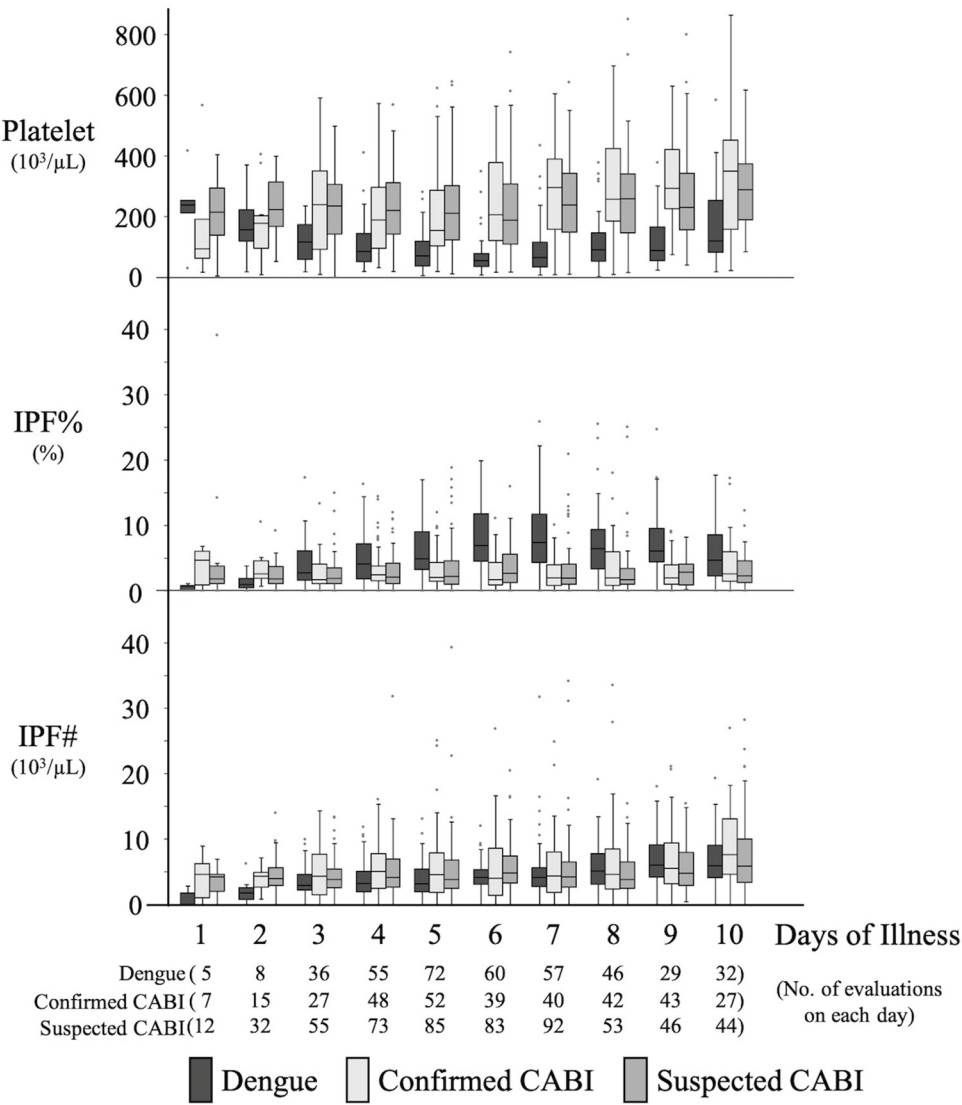

**Fig 2. Time course trends of platelet parameters in each group from the first to tenth days of illness.** Box and whisker plots show the time course trends of platelet parameters in each group from the first to tenth days of illness. Boxes show the median and interquartile values, whiskers represent the upper and lower adjacent values and dots indicate outside values. The total numbers of evaluations included on each day are indicated on the X axis by each group. CABI: community acquired bacterial infection, IPF%: Immature Platelet Fraction, IPF#: Immature Platelet Fraction Count.

increasing trend although the dengue group was slightly steeper. The line plot of the same data is shown in S2 Fig. The time courses of the platelet parameters in the dengue group are shown by age groups in S3 Fig and their trends were similar.

## 4. Comparison of platelet parameters among dengue, confirmed CABI, and suspected CABI groups by specific time-phases

Fig 3 shows the comparison of platelet parameters among dengue, confirmed CABI, and suspected CABI groups by three time-phases; febrile phase (day 1 to 3), critical phase (day 4 to 6), and recovery phase (day 7 to 10). (A) shows a comparison including all participants. The

platelet counts of the dengue group were significantly lower than those of both the confirmed and suspected CABI groups in all phases. There was no significant difference of IPF% between dengue group and confirmed CABI group in the febrile phase (median 1.9% versus 2.4%; p = 0.488), however, the IPF% of dengue group increased to be significantly higher than that of the confirmed CABI in the critical phase (median 5.2% versus 2.2%; p <0.001) and recovery phase (median 6.5% versus 2.2%; p <0.001). The IPF% of the dengue group was also significantly higher than that of the suspected CABI groups in the critical and recovery phases. The IPF# of the dengue group was significantly lower than those of the confirmed and suspected CABI groups in the febrile and critical phases. S4 Fig shows a comparison including only the subgroups with non-severe thrombocytopenia and significant differences of IPF% and IPF# were also shown between dengue and the CABI groups in similar phases. (B) shows a comparison including only the subgroups with severe thrombocytopenia (defined as platelet nadir $<50\times10^3$/μl) during hospitalization. There were 67 dengue, 21 confirmed CABI and 25 suspected CABI subjects with severe thrombocytopenia. The IPF% of the dengue group remained significantly higher than that of the confirmed CABI group in the critical phase (median 8.2% versus 2.0%; p <0.001) and recovery phases (median 9.2% versus 5.3%; p = 0.02) even though there was no significant difference of platelet counts between the groups in the same phases. The IPF% of the dengue group also remained significantly higher than that of the suspected CABI group in the recovery phases. The IPF# of the confirmed CABI was significantly lower than that of the dengue group in the critical phase despite no significant difference of platelet counts between the groups in that same phase.

## 5. Comparison of platelet parameters between the severe thrombocytopenia dengue group and the non-severe thrombocytopenia dengue group by specific time-phases

Fig 4 shows the comparison of platelet parameters between the severe thrombocytopenia dengue group defined as platelet nadir $<50\times10^3$/μl during hospitalization and the non-severe thrombocytopenia dengue group defined to include all patients with a platelet nadir $\geq50\times10^3$/μl during hospitalization by specific time-phases. The platelet counts of the severe thrombocytopenia dengue group were significantly lower than those of the non-severe thrombocytopenia dengue group in all phases. The IPF% of the severe thrombocytopenia dengue group was significantly higher than that of the non-severe thrombocytopenia dengue group in the critical phase (median 8.2% versus 3.8%; p <0.001) and recovery phases (median 9.2% versus 4.3%; p <0.001). The IPF# of the severe thrombocytopenia dengue group was significantly lower than that of the non-severe thrombocytopenia dengue group only in the critical phases. S5 Fig shows a comparison between the severe and non-severe dengue groups (defined in the "Disease definition" section) There were 50 severe dengue subjects. For all platelet parameters there was no significant difference between the severe dengue and non-severe dengue groups in all phases.

## Discussion

In this study we investigated the platelet parameters IPF% and IPF# in a dengue group at the point of admission and during the course of hospitalization. Our evaluation revealed unique platelet pattern parameters in the dengue group when compared with those of the confirmed and suspected CABI groups. In brief, in a time course evaluation the IPF% of the patients with dengue infection was significantly higher than those of the confirmed and suspected CABI groups at admission and during the critical and recovery phases. When we considered only those with severe thrombocytopenia, the IPF% of the dengue group remained significantly

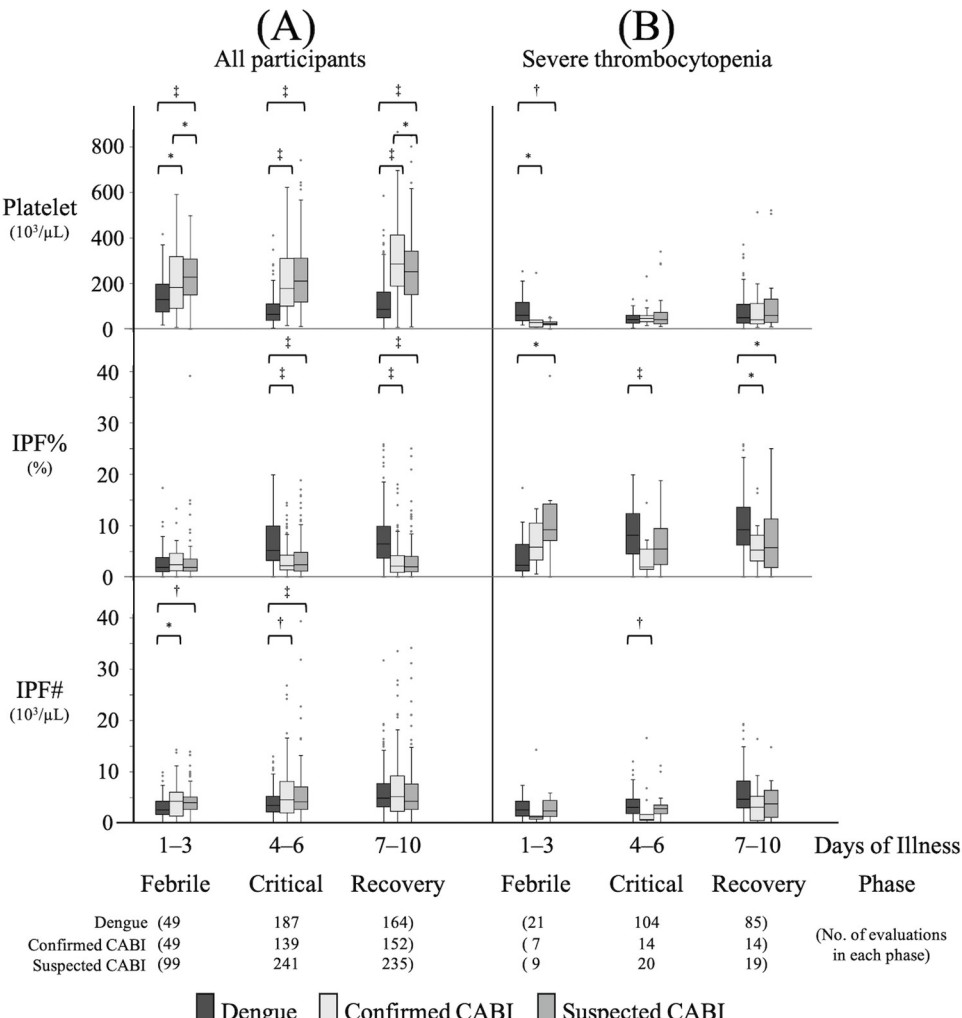

**Fig 3. Comparison of platelet parameters among dengue, confirmed CABI, and suspected CABI groups by specific time-phases.** Box and whisker plots show the platelet parameters of dengue, confirmed CABI, and suspected CABI groups observed in specific time-phases: febrile phase (day 1 to 3), critical phase (day 4 to 6), and recovery phase (day 7 to 10). (A) shows a comparison including all participants and (B) shows a comparison including only the subgroups with severe thrombocytopenia defined as platelet nadir $<50\times10^3/\mu l$ during hospitalization. Boxes show the median and interquartile values, whiskers represent the upper and lower adjacent values and dots indicate outside values. Comparison of dengue, confirmed CABI, and suspected CABI groups were performed using the Kruskal-Wallis test followed by the Dunn's post hoc test with Holm adjustment. *: P<0.05, †: P<0.01, ‡: P<0.001. CABI: community acquired bacterial infection, IPF%: Immature Platelet Fraction, IPF#: Immature Platelet Fraction Count.

higher than those of the confirmed (at admission, and in the critical and recovery phases) and suspected CABI groups (only in the recovery phase) even though their platelet levels were comparable in the same phases. Similarly, the IPF# of the confirmed CABI group was lower than that of the dengue group at admission and in the critical phase, with matching severity of thrombocytopenia.

It is assumed that an increased IPF% indicates a consumptive or destructive thrombocytopenic status and a normal or decreased IPF% suggests decreased platelet production in bone marrow. IPF# is assumed to reflect real-time platelet production [7, 18, 19]. However, one study claimed that IPF# changes should be approached with caution because fluctuations might be diminished by converting IPF% to absolute numbers in thrombocytopenia [20].

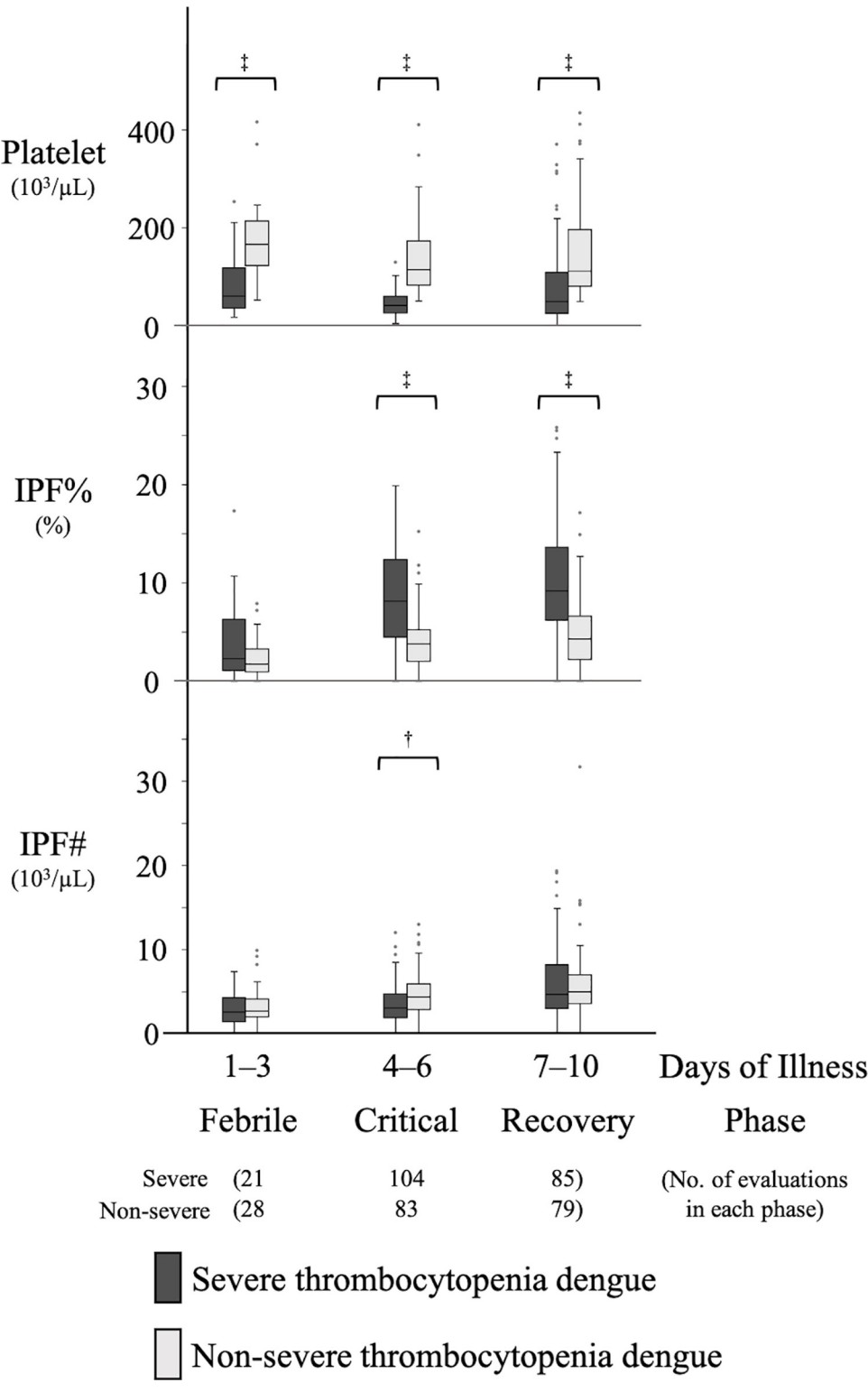

**Fig 4. Comparison of platelet parameters between the severe and non-severe dengue thrombocytopenia groups by specific time-phases.** Box and whisker plots show the platelet parameters of the severe thrombocytopenia dengue group defined as platelet nadir $<50\times10^{3}/\mu l$ during hospitalization and the non-severe thrombocytopenia dengue group defined to include all patients with a platelet nadir $\geq50\times10^{3}/\mu l$ during hospitalization observed in specific time-phases: febrile phase (day 1 to 3), critical phase (day 4 to 6), and recovery phase (day 7 to 10). Boxes show the median and interquartile values, whiskers represent the upper and lower adjacent values and dots indicate outside values. The

comparison between each pair of dengue subgroups were performed using the Mann Whitney test. *: P<0.05, †: P<0.01, ‡: P<0.001. IPF%: Immature Platelet Fraction, IPF#: Immature Platelet Fraction Count.

Moreover, immature platelets are also susceptible to consumption or destruction, which may lower IPF# if the rate of platelet loss exceeds the pace of platelet production. Clinical interpretations of IPF% and IPF# have been established based on observations mainly among hematologic disorders. For example, high IPF% was reported in ITP and TTP, which are caused by increased platelet consumption, while low to normal IPF% and low IPF# were documented in aplastic anemia and chemotherapy-induced thrombocytopenia, which are caused by decreased platelet production in bone marrow [6, 9, 11]. Low IPF# was observed in ITP with high IPF%, which is compatible with the fact that ITP is a multifactorial autoimmune disease characterized by both increased platelet destruction and/or reduced platelet production; and increased IPF# was observed in ITP when treated by a thrombopoietin receptor agonist which increases platelet production [7, 9]. These observations support clinical interpretations of IPF% and IPF#. Recently, IPF% and IPF# were further applied for infectious syndromes such as sepsis [21–23].

Hypotheses proposed regarding the underlying mechanisms of dengue-induced thrombocytopenia are categorized mainly into two: decreased platelet production by bone marrow or increased peripheral platelet consumption or destruction. Some previous studies proposed the infection of hematopoietic progenitors or stromal cells as the causes of dengue-induced decreased platelet production by bone marrow, and other studies proposed the anti-platelet autoantibodies, the platelet-endothelial interaction, the platelet-leukocyte interaction, the platelet-virus interaction or the soluble factors as the causes of dengue-induced increased peripheral platelet consumption/destruction [24, 25]. In this study the IPF% of the dengue group was significantly higher than those of confirmed and suspected CABI groups in critical and/or recovery phases even after matching thrombocytopenia levels. Additionally, comparison between the severe thrombocytopenia and non-severe thrombocytopenia dengue groups showed a significantly higher IPF% of the severe thrombocytopenia dengue group versus those of other groups in the critical and recovery phases. Considering that IPF% reflects the balance of two potential mechanisms—increased platelet consumption or destruction with high IPF% and decreased platelet production with normal or low IPF%—these results suggest the predominance of increased peripheral platelet consumption or destruction over decreased platelet production by bone marrow as an underlying mechanism of dengue-induced thrombocytopenia in the critical and recovery phases. Moreover, decreased platelet production by bone marrow might potentially co-exist with peripheral platelet consumption or destruction at the peak of dengue-induced thrombocytopenia, because the comparison between the severe thrombocytopenia dengue and the non-severe thrombocytopenia dengue groups showed a significantly lower IPF# of the severe thrombocytopenia group versus those of other groups in the critical phase. However, the IPF# need to be interpreted with caution because of the susceptivity of immature platelets to consumption or destruction as mentioned above. Because the IPF% and the IPF# of the confirmed CABI group were significantly lower than that of dengue in the critical phase after matching the thrombocytopenia levels, the suppression of platelet production by bone marrow seems to be more predominant in CABI-induced than in dengue-induced thrombocytopenia.

Drawing from their utility in hematologic disorders, IPF% and IPF# offer insight into dengue-induced thrombocytopenia as well as provide information for the clinical management of dengue patients. Our results show differences of IPF% and IPF# between dengue and the CABI groups. IPF% and IPF# are potentially valuable parameters for the differential diagnosis of dengue from CABI especially in patients with thrombocytopenia. Further study is warranted

to analyze the performance of IPF as a biomarker for the differential diagnosis of dengue from CABI.

This study has some limitations. Sequential day-by-day test results were not available for all patients, and this may introduce overestimation when severe patients were tested more frequently. There may be unreliability in patient reported onset day of symptoms. Because of the limited diagnostic techniques available at the study site, few varieties of bacterial infectious diseases could be included as confirmed CABI. Not all pneumonia-suspected patients underwent X-ray tests at the study site because of the limited resources. We rigorously judged the existence of radiographic shadowing for high specificity, which might reduce the number of X-ray confirmed pneumonia cases. It was difficult to include confirmed urinary tract infections because a clean catch urine collection was technically difficult and bacterial contamination occurred very frequently at the study site. Detailed investigations were not available at the study site to determine possible primary or secondary dengue infections and dengue serotypes. We could not check serum thrombopoietin, the main factor for regulating thrombopoiesis. Although disseminated intravascular coagulation (DIC) might influence the IPF results, the coagulation studies to assess DIC were not available at the study site. This was a single-center study and further data accumulation is required in various countries and clinical settings to validate our findings.

This study demonstrated the distinctive characteristics and time-course trends of IPF% and IPF# in a dengue group, which were significantly different compared with those of confirmed and suspected CABI groups. IPF% and IPF# are potentially valuable parameters in dengue and further investigation is required for the optimal use in clinical practice.

## Supporting information

**S1 Fig. Comparison of platelet, IPF%, and IPF# among dengue and each confirmed CABI at admission.** The lines show the median with interquartile ranges. Comparisons of each group were performed using the Kruskal-Wallis test followed by the Dunn's post hoc test with Holm adjustment. *: $P<0.05$, †: $P<0.01$, ‡: $P<0.001$. IPF%: Immature Platelet Fraction, IPF#: Immature Platelet Fraction Count.
(TIF)

**S2 Fig. Time course trends of platelet parameters in each group from the first to tenth days of illness (line plot).** Data are expressed as mean with a 95% confidence interval shown by the error bars. The total number of evaluations included on each day is indicated on the X axis by each group. The mean of each platelet parameter by days of illness were calculated from the results of participants who underwent a blood test on each day. CABI: community acquired bacterial infection, IPF%: Immature Platelet Fraction, IPF#: Immature Platelet Fraction Count.
(TIF)

**S3 Fig. Time course of platelet parameters in the dengue group by age groups.** Box and whisker plots show the time course trends of platelet parameters in the dengue group by age from the first to tenth days of illness. Boxes show the median and interquartile values, whiskers represent upper and lower adjacent values and dots indicate outside values. The total number of evaluations included on each day is indicated on the X axis by each group. IPF%: Immature Platelet Fraction, IPF#: Immature Platelet Fraction Count.
(TIF)

**S4 Fig. Comparison of platelet parameters among dengue, confirmed CABI, and suspected CABI groups including only the subgroups with non-severe thrombocytopenia by specific**

**time-phases.** Box and whisker plots show the platelet parameters of dengue, confirmed CABI, and suspected CABI groups observed in specific time-phases: febrile phase (day 1 to 3), critical phase (day 4 to 6), and recovery phase (day 7 to 10). Non-severe thrombocytopenia was defined as a platelet nadir $\geq 50 \times 10^3$/μl during hospitalization. Boxes show the median and interquartile values, whiskers represent the upper and lower adjacent values and dots indicate outside values. Comparison of dengue, confirmed CABI, and suspected CABI groups were performed using the Kruskal-Wallis test followed by the Dunn's post hoc test with Holm adjustment. *: P<0.05, †: P<0.01, ‡: P<0.001. CABI: community acquired bacterial infection, IPF%: Immature Platelet Fraction, IPF#: Immature Platelet Fraction Count.
(TIF)

**S5 Fig. Comparison of platelet parameters between the severe and non-severe dengue groups by specific time-phases.** Box and whisker plots show the platelet parameters of severe and non-severe dengue groups observed in specific time-phases; febrile phase (day 1 to 3), critical phase (day 4 to 6) and recovery phase (day 7 to 10). Severe dengue was clinically defined in the "Disease definition" section. Boxes show the median and interquartile values, whiskers represent the upper and lower adjacent values and dots indicate outside values. Comparisons between each pair of dengue subgroups were performed using the Mann Whitney test. IPF%: Immature Platelet Fraction, IPF#: Immature Platelet Fraction Count.
(TIF)

**S1 Table. Comparison of the results from real-time PCR (*rrs* real-time PCR) and *flaB*-nested PCR.** A total number of 50 negative samples and 43 positive samples for *flaB*-nested PCR were tested in duplicate by *rrs* real-time PCR. The 50 negative samples for *flaB*-nested PCR were also negative for the microscopic agglutination test using paired sera. This *rrs* real-time PCR did not detect bacterial species other than *Leptospira* spp. described in published study [17]. *rrs*: 16S ribosomal RNA gene.
(DOCX)

# Acknowledgments

We thank all the patients who participated in this study. We are grateful to the Director and staff of San Lazaro Hospital for their support in the conduct of this study. We thank Christopher M. Parry for commenting on previous drafts. We also wish to thank SLH-Nagasaki research staffs.

# Author Contributions

**Conceptualization:** Nobuo Saito, Motoi Suzuki, Koya Ariyoshi.

**Data curation:** Ikkoh Yasuda, Nobuo Saito.

**Formal analysis:** Ikkoh Yasuda, Nobuo Saito, Chris Fook Sheng Ng.

**Funding acquisition:** Nobuo Saito, Koya Ariyoshi.

**Investigation:** Ikkoh Yasuda, Nobuo Saito, Dorcas Valencia Umipig, Rontgene M. Solante, Ferdinand De Guzman, Ana Ria Sayo, Kentaro Sakashita.

**Methodology:** Nobuo Saito, Motoi Suzuki, Michio Yasunami, Nobuo Koizumi, Emi Kitashoji, Koya Ariyoshi.

**Project administration:** Nobuo Saito.

**Resources:** Ikkoh Yasuda, Nobuo Saito, Dorcas Valencia Umipig, Rontgene M. Solante, Ferdinand De Guzman, Ana Ria Sayo, Michio Yasunami, Nobuo Koizumi, Emi Kitashoji, Kentaro Sakashita.

**Software:** Ikkoh Yasuda, Nobuo Saito.

**Supervision:** Nobuo Saito, Motoi Suzuki, Koya Ariyoshi.

**Validation:** Ikkoh Yasuda, Nobuo Saito.

**Visualization:** Ikkoh Yasuda, Nobuo Saito.

**Writing – original draft:** Ikkoh Yasuda.

**Writing – review & editing:** Nobuo Saito, Chris Fook Sheng Ng, Chris Smith, Koya Ariyoshi.

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
