## [Decision Letter · Decision Letter 0]

2 Aug 2021

PONE-D-21-17692

Unique characteristics of new CBC parameters, the Immature Platelet Fraction and the Immature Platelet Fraction Count, in dengue patients.

PLOS ONE

Dear Dr. Yasuda,

Thank you for submitting your manuscript to PLOS ONE. After careful consideration, we feel that it has merit but does not fully meet PLOS ONE’s publication criteria as it currently stands. Therefore, we invite you to submit a revised version of the manuscript that addresses the points raised during the review process.

We look forward to receiving your revised manuscript.

Kind regards,

Elizabeth S. Mayne, M.D.

Academic Editor

PLOS ONE

Journal Requirements:

2. Please include information in the Ethics statement as to whether the IRB approved the consent procedures. Please clarify if any consent or assent was provided by patients for whom guardians provided written consent.

4. Thank you for stating the following in the Competing Interests/Financial Disclosure section: 

I have read the journal's policy and the authors of this manuscript have the following competing interests: School of Tropical Medicine and Global Health, Nagasaki University was supplied with an automated hematology analyzer, Sysmex XN-1000, and the related reagents from Sysmex Corporation for the performance of this study. Each author has no competing interests individually. 

We note that you received funding from a commercial source: Sysmex Corporation

Additional Editor Comments:

This is an interesting study looking at full blood count parameters in Dengue fever. Although the manuscript has a number of valuable points, both reviewers felt that there were revisions that were required. The most important of these is the comparison of those patients with Dengue fever (a viral infection) to patients with community acquired bacterial infections. If this is an appropriate control population as suggested, then this should be justified in the introduction. One reviewer felt that a comparison with a normal healthy population (or population specific reference ranges) may be more appropriate. Both reviewers were concerned with the group that constituted "assumed CABI" especially given that some of these had pneumonia without X-ray confirmation.

The white cell results should ideally be represented as numerical values and as percentages and I agree with reviewer 1 that exclusion of a disseminated intravascular coagulation would be of value here.

Reviewers' comments:

Reviewer's Responses to Questions

**Comments to the Author**

1. Is the manuscript technically sound, and do the data support the conclusions?

Reviewer #1: Yes

Reviewer #2: Partly

2. Has the statistical analysis been performed appropriately and rigorously? 

Reviewer #1: Yes

Reviewer #2: No

3. Have the authors made all data underlying the findings in their manuscript fully available?

Reviewer #1: Yes

Reviewer #2: Yes

4. Is the manuscript presented in an intelligible fashion and written in standard English?

Reviewer #1: Yes

Reviewer #2: Yes

5. Review Comments to the Author

Reviewer #1: This manuscript describes the characteristics of the IPF% and IPF# over time in patients with dengue fever. This is of interest, as it contributes to our knowledge regarding the mechanistic cause for thrombocytopenia in dengue. The paper is generally well written, but requires some modification:

1) The introduction would benefit from some brief background information about dengue.

2) The suspected pathophysiological mechanisms of thrombocytopenia mentioned in line 76 and 77 of the introduction should be expanded upon.

3) There is a minor typographical error in line 87 (IPF# written as IFP#).

4) In line 105, it is stated that the “confirmed CABI group was potentially weighted toward bacterial infection”. This does not make sense, as CABI was by definition “community acquired bacterial infection”.

5) The CABI disease definition criteria are unclear. Presumably they included Bacteraemia, Diptheria, Meningococcal disease, leptospirosis, X-Ray confirmed pneumonia and skin infection?

6) Suspected CABI is defined as cases where a diagnosis of confirmed CABI could not be made. However, the suspected CABI cases listed in lines 199-202 seem to include CABI-defining conditions? Did 72 patients have pneumonia not confirmed on X-Ray? There are also 10 cases of Leptospirosis and 1 of meningococcal disease? Why were there no patients with urinary tract infections included in the CABI cases? Surely some must have had confirmed bacterial infection?

7) In line 131-132, severe thrombocytopenia is defined, but non-severe thrombocytopenia is not. Did the latter include patients with thrombocytopenia with platelet counts >50x1-^9/l, or all patients with platelets >50x10^9/l (with and without thrombocytopenia)?

8) It would be more meaningful to present the results of the differential counts (Neutrophils, Lymphocytes, monocytes and Eosinophils) in Table 1 as absolute numbers instead of percentages.

9) As you have patients of varying ages and genders, comparison of the median Hb is potentially problematic (as reference intervals vary quite substantially according to age and gender for this parameter). I would suggest including the percentage of patients with anaemia in this table.

10) As a consumptive coagulopathy (viz. DIC) may contribute to the thrombocytopenia in patients with dengue, results of coagulation studies (Prothrombin time, PTT, Fibrinogen +/- D-Dimers) should ideally be included in Table 1, including the proportion with laboratory evidence of a coagulopathy. The contribution of a coagulopathy to the IPF results should also be assessed and discussed if possible.

11) The abbreviations used in Table 1 should be defined in the table’s legend (BUN, ALT, AST, etc).

12) In line 384 of the discussion, it is stated that the findings “might suggest the predominance of increased peripheral platelet consumption”. This statement is very vague. The findings do suggest the predominance of increased peripheral platelet consumption in dengue-induced thrombocytopenia.

13) The conclusion drawn in line 389-392 regarding the rate of platelet production by the bone marrow must be couched with some caution, as immature platelets are also susceptible to consumption or destruction, which may lower their absolute count if the rate of platelet loss exceeds the pace of platelet production.

14) In line 392, include the IPF% with the IPF# to support the depressed platelet production in CABI with thrombocytopenia.

15) In line 401-402, it is stated that the parameters (IPF% abd IPF#) are potentially valuable for the differential diagnosis of dengue vs CABI. This should be clarified to be in the differential diagnosis of dengue vs CABI in patients with thrombocytopenia (as the IPF data in non-severe thrombocytopenia patients looks fairly similar to the CABI patients (Fig. 3 vs Fig. 2A)? Further analysis in this regard may be of value to further assess the potential for the IPF to discriminate dengue from CABI/suspected CABI in patients without severe thrombocytopenia.

16) The statement from line 402-403 that the parameters (IPF% and IPF#) could be used to predict severity of dengue-induced thrombocytopenia and anticipation of platelet recovery seems unsupported by the data.

Reviewer #2: The study provides information that is locally relevant on the applicability of novel parameters that would be routinely available and adds to the understanding of the pathophysiology of the disease process. This information could possible serve to risk stratify patients in resource poor environments.

6. PLOS authors have the option to publish the peer review history of their article (what does this mean?). If published, this will include your full peer review and any attached files.

Reviewer #1: No

Reviewer #2: No

---

## [Author Response · Author response to Decision Letter 0]

9 Sep 2021

Editor

“This is an interesting study looking at full blood count parameters in Dengue fever. Although the manuscript has a number of valuable points, both reviewers felt that there were revisions that were required.”

We are very thankful for your thorough review. We have incorporated changes that reflect the detailed suggestions you have kindly provided.

Comments:

[Comment 1] Please ensure that your manuscript meets PLOS ONE's style requirements, including those for file naming.

Response: The following are the points revised in accordance with the PLoS ONE’s style requirement. (The parts with underline)

Line 107 (these line numbers correspond to the line numbers in the manuscript without tracked changes):

“Materials and methods”

Line 178:

“Ethical issues”

[Comment 2] Please include information in the Ethics statement as to whether the IRB approved the consent procedures. Please clarify if any consent or assent was provided by patients for whom guardians provided written consent.

Response: The IRBs approved the consent procedures of our study. The requirements of the IRBs during the study period included obtaining written informed consent; however, they did not include obtaining assent of the patients for whom guardians provided written consent. We tried to verbally explain the study following the consent form to these patients if they were able to understand it. To clarify this point, we added text as follows.

Lines 186-189:

“The requirements of the institutional review boards during the study period included obtaining written informed consent; however, they did not include obtaining assent of the patients for whom guardians or caregivers provided written consent.”

Line 189-190:

“The institutional review boards approved the consent procedures.”

[Comment 3] Please know it is PLOS ONE policy for corresponding authors to declare, on behalf of all authors, all potential competing interests for the purposes of transparency.

Response: We declared all potential competing interests in the “Competing Interests/Financial Disclosure” section.

[Comment 4] Please provide an amended Competing Interests Statement that explicitly states this commercial funder, along with any other relevant declarations relating to employment, consultancy, patents, products in development, marketed products, etc.” “Within this Competing Interests Statement, please confirm that this does not alter your adherence to all PLOS ONE policies on sharing data and materials by including the following statement: "This does not alter our adherence to PLOS ONE policies on sharing data and materials.”” “Please include your amended Competing Interests Statement within your cover letter. We will change the online submission form on your behalf.

Response: We have amended the Competing Interests Statement to explicitly states the commercial funder, Sysmex Corporation, as below. We would appreciate if you could change the online submission form to the new one.

“I have read the journal's policy and the authors of this manuscript have the following competing interests: The School of Tropical Medicine and Global Health, Nagasaki University was supplied with an automated hematology analyzer, Sysmex XN-1000, and the related reagents from Sysmex Corporation for the performance of this study. The authors have no conflicts of interest associated with Sysmex Corporation relating to the employment, consultancy, patents, products in development, and marketed products. This does not alter our adherence to PLoS ONE policies on sharing data and materials. Each author has no competing interests individually.”

[Comment 5] Although the manuscript has a number of valuable points, both reviewers felt that there were revisions that were required. The most important of these is the comparison of those patients with Dengue fever (a viral infection) to patients with community acquired bacterial infections. If this is an appropriate control population as suggested, then this should be justified in the introduction. One reviewer felt that a comparison with a normal healthy population (or population specific reference ranges) may be more appropriate.

Response: Thank you for an important question. First, the main topic of this study was the utility of IPF among febrile patients with thrombocytopenia. Second, it is valuable to assess IPF% particularly among patients with thrombocytopenia because IPF% is a parameter representing the ratio of the absolute number of immature platelets to the total number of platelets. For these purposes, we believe that febrile infectious diseases that can induce thrombocytopenia should be a control group. We chose community acquired bacterial infection (CABI) as a control group because CABI can induce thrombocytopenia the same as dengue and it has clinically important differences from dengue; specifically, the need for antibiotic treatment and poorer prognosis when accompanied by thrombocytopenia. By comparing dengue and CABI, this study could indicate different mechanisms of thrombocytopenia, each different response of thrombopoietic activity to thrombocytopenia, and the potential of IPF for the differential diagnosis of dengue from CABI, which would not have been achieved by a comparison with a normal healthy population. We have added the text to reflect these points as follows.

Lines 99-104:

“CABI was chosen as the control group because the main topic of this study was the utility of IPF among febrile patients with thrombocytopenia. CABI can induce thrombocytopenia the same as dengue and it has clinically significant differences from dengue; specifically, the need for antibiotic treatment and poorer prognosis when accompanied by thrombocytopenia.”

[Comment 6] Both reviewers were concerned with the group that constituted "assumed CABI" especially given that some of these had pneumonia without X-ray confirmation.

Response: The pneumonia cases of the suspected CABI group were clinically suspected as pneumonia; however, they did not undergo an X-ray test or their radiographic shadowing in lung fields were not obvious. Because of the limited resources, not all pneumonia-suspected patients underwent X-ray tests at the study site, especially if they were mild or pediatric cases. We rigorously judged the existence of radiographic shadowing for high specificity, which might reduce the number of confirmed pneumonia cases (regarding other diseases of suspected CABI, please see our responses to the comment 5 and 6 of reviewer #1) We recognize this point as a limitation of this study. We have described this limitation as follows.

Lines 432-435:

“Not all pneumonia-suspected patients underwent X-ray tests at the study site because of the limited resources. We rigorously judged the existence of radiographic shadowing for high specificity, which might reduce the number of X-ray confirmed pneumonia cases.

[Comment 7] The white cell results should ideally be represented as numerical values and as percentages

Response: Thank you for your suggestion. We have revised the Table to represent the white cell results both as numerical values and as percentages. (Line 213)

[Comment 8] I agree with reviewer 1 that exclusion of a disseminated intravascular coagulation would be of value here.

Response: We agree that disseminated intravascular coagulation might influence the IPF results. However, the coagulation studies were not available at the study site, and we could not exclude patients with disseminated intravascular coagulation. We have clearly described this limitation as follows.

Lines 440-442:

“Although disseminated intravascular coagulation (DIC) might influence the IPF results, the coagulation studies to assess DIC were not available at the study site.”

Reviewer #1

This manuscript describes the characteristics of the IPF% and IPF# over time in patients with dengue fever. This is of interest, as it contributes to our knowledge regarding the mechanistic cause for thrombocytopenia in dengue. The paper is generally well written, but requires some modification

Thank you for the thoughtful and constructive feedback you provided regarding our manuscript, and we are confident that you will find that this most recent version of our manuscript clears up the main issues raised.

Comments:

[Comment 1] The introduction would benefit from some brief background information about dengue.

Response: We have added text with brief background information of dengue in the Introduction section, as below:

Lines 77-80 (these line numbers correspond to the line numbers in the manuscript without tracked changes):

“Dengue is a mosquito-borne viral infection and numerically one of the most important viral diseases in tropical area. [1,2] A total of 390 million dengue infections are estimated per year, and 3.97 billion people are estimated to be at risk of dengue infection worldwide. [3,4]”

[Comment 2] The suspected pathophysiological mechanisms of thrombocytopenia mentioned in line 76 and 77 of the introduction should be expanded upon.

Response: We have included new text to describe the details of the suspected pathophysiological mechanisms of dengue-induced thrombocytopenia in the Discussion section as below:

Lines 393-398:

“Some previous studies proposed the infection of hematopoietic progenitors or stromal cells as the causes of dengue-induced decreased platelet production by bone marrow, and other studies proposed the anti-platelet autoantibodies, the platelet-endothelial interaction, the platelet-leukocyte interaction, the platelet-virus interaction or the soluble factors as the causes of dengue-induced increased peripheral platelet consumption/destruction.”

[Comment 3] There is a minor typographical error in line 87 (IPF# written as IFP#).

Response: Thank you very much for letting us know. The typographical error has been corrected as below.

Lines 94-95:

“Although IPF% and IPF# have been implemented in wider clinical settings,”

[Comment 4] In line 105, it is stated that the “confirmed CABI group was potentially weighted toward bacterial infection”. This does not make sense, as CABI was by definition “community acquired bacterial infection.”

Response: In this revision this section is clarified as follows. Because limited diagnostic techniques were available at the study site, few varieties of bacterial infectious disease were definitively diagnosable. If we included such a definitive diagnostic of disease, then a selection bias would be introduced. Therefore, we also included a “suspected CABI” category. We have rewritten the text as below to clarify this point.

Lines 116-121:

“Because the limited diagnostic techniques were available at the study site and few varieties of bacterial infectious diseases could be definitively diagnosed, the confirmed CABI group was potentially weighted toward the diseases that were definitively diagnosable at the study site. Therefore, the suspected CABI was included as a category to avoid selection bias.”

[Comment 5] The CABI disease definition criteria are unclear. Presumably they included Bacteraemia, Diptheria, Meningococcal disease, leptospirosis, X-Ray confirmed pneumonia and skin infection?

Response: Thank you very much for this comment. As mentioned above (please see our response to the comment 4 of reviewer #1), because of the limited diagnostic techniques available at the study site, there were few bacterial infectious diseases that we could diagnose definitively. The “definitively diagnosable diseases at the study site” were bacteraemia, diptheria, meningococcal disease, leptospirosis, X-Ray confirmed pneumonia, and skin infection. The definition criteria are described in S1 Appendix (we have moved the definition criteria of CABIs from the previous “Disease definition” section to S1 Appendix). The CABI that fulfilled these definition criteria of CABI were categorized as confirmed CABIs. We agree that the limited diagnostic techniques were a limitation of this study. We have added new text to clarify this point as follows:

Lines 132-134:

“The CABI that fulfilled the diagnostic criteria was categorize as confirmed CABI. Suspected CABI was a clinical diagnosis after excluding dengue infection but unable to reach a confirmed CABI diagnosis described in S1 appendix.”

Lines 431-432:

“Because of the limited diagnostic techniques available at the study site, few varieties of bacterial infectious diseases could be included as confirmed CABI.”

[Comment 6] Suspected CABI is defined as cases where a diagnosis of confirmed CABI could not be made. However, the suspected CABI cases listed in lines 199-202 seem to include CABI-defining conditions? Did 72 patients have pneumonia not confirmed on X-Ray? There are also 10 cases of Leptospirosis and 1 of meningococcal disease? Why were there no patients with urinary tract infections included in the CABI cases? Surely some must have had confirmed bacterial infection?

Response: We agree that these points were not clear in the original manuscript, and in revision have attempted to clarify All the suspected CABI cases listed in lines 199-202 of the previous manuscript were diagnosed clinically, because they were unable to reach a confirmed CABI diagnosis. For example, the case of meningococcal disease of the suspected CABI group was the case that was clinically diagnosed as meningococcal disease; however, the case was unable to fulfill the diagnostic criteria of meningococcal disease described in S1 Appendix (we have moved the definition criteria of CABIs from the previous “Disease definition” section to S1 Appendix). This suspected meningococcal disease case could not provide a positive result from neither culture nor PCR test for meningococcus. Regarding the 72 pneumonia cases in the suspected CABI group, they were cases that were clinically suspected as pneumonia; however, they did not undergo an X-ray test or their radiographic shadowing in lung fields were not obvious. Because of the limited resources, not all pneumonia-suspected patients underwent X-ray tests at the study site especially if they are mild or pediatric cases . Moreover, we rigorously judged the existence of radiographic shadowing for high specificity, which might reduce the number of X-ray confirmed pneumonia cases. Regarding urinary tract infections, it was difficult to include confirmed urinary tract infections because a clean catch urine collection was technically difficult and bacterial contamination occurred frequently at the study site. We recognize the lack of confirmed urinary tract infection as one of the important limitations of this study. We have added the following text to reflect these points:

Lines 207-208:

“By definition, all suspected CABIs were diagnosed clinically and no suspected CABI patient fulfilled the diagnostic criteria of confirmed CABIs.”

Lines 432-438:

“Not all pneumonia-suspected patients underwent X-ray tests at the study site because of the limited resources. We rigorously judged the existence of radiographic shadowing for high specificity, which might reduce the number of X-ray confirmed pneumonia cases. It was difficult to include confirmed urinary tract infections because a clean catch urine collection was technically difficult and bacterial contamination occurred very frequently at the study site.”

[Comment 7] In line 131-132, severe thrombocytopenia is defined, but non-severe thrombocytopenia is not. Did the latter include patients with thrombocytopenia with platelet counts >50x1-^9/l, or all patients with platelets >50x10^9/l (with and without thrombocytopenia)?

Response: Thank you very much for this important comment. In revision we have clarified that the “non-severe thrombocytopenia” in the text includes all patients with platelets ≥50×103/µl regardless of whether they have thrombocytopenia or not, as follows:

Lines 136-139:

“Non-severe thrombocytopenia was defined to include all patients with platelets ≥50×103/µl at admission or a platelet nadir ≥50×103/µl during hospitalization in a time course evaluation.”

Lines 329-333:

“Fig 4 shows the comparison of platelet parameters between the severe thrombocytopenia dengue group defined as platelet nadir <50×103/µl during hospitalization and the non-severe thrombocytopenia dengue group defined to include all patients with a platelet nadir ≥50×103/µl during hospitalization by specific time-phases.”

Lines 347-350:

“Box and whisker plots show the platelet parameters of the severe thrombocytopenia dengue group defined as platelet nadir <50×103/µl during hospitalization and the non-severe thrombocytopenia dengue group defined to include all patients with a platelet nadir ≥50×103/µl during hospitalization observed in specific time-phases …”

[Comment 8] It would be more meaningful to present the results of the differential counts (Neutrophils, Lymphocytes, monocytes and Eosinophils) in Table 1 as absolute numbers instead of percentages.

Response: Thank you, the Editor also advised to show the differential counts both as numerical values and as percentages. We have revised the Table to represent the white cell results both as numerical values and as percentages. (Line 213)

[Comment 9] As you have patients of varying ages and genders, comparison of the median Hb is potentially problematic (as reference intervals vary quite substantially according to age and gender for this parameter). I would suggest including the percentage of patients with anaemia in this table.

Response: We agree, and in revision have included the percentage of patients with anemia at admission in the table and added the texts regarding the definition of anemia in the “Disease definition” section as below.

Lines 213:

“(In Table) Patients with anemia at admission: 23 (15.1), 93 (51.7), <0.001, 94 (34.3), <0.001”

Lines 143-144:

“Anemia was defined by hemoglobin levels at admission according to WHO guidelines. [12]”

Lines 495-497:

“12. World Health Organization. Haemoglobin concentrations for the diagnosis of anaemia and assessment of severity 2011. Available from: https://apps.who.int/iris/handle/10665/85839.”

[Comment 10] As a consumptive coagulopathy (viz. DIC) may contribute to the thrombocytopenia in patients with dengue, results of coagulation studies (Prothrombin time, PTT, Fibrinogen +/- D-Dimers) should ideally be included in Table 1, including the proportion with laboratory evidence of a coagulopathy. The contribution of a coagulopathy to the IPF results should also be assessed and discussed if possible.

Response: We agree that disseminated intravascular coagulation may contribute to the thrombocytopenia and have influence on the IPF results. However, the coagulation studies were not available at the study site. We have described this limitation as follows.

Lines 440-442:

“Although disseminated intravascular coagulation (DIC) might influence the IPF results, the coagulation studies to assess DIC were not available at the study site.”

[Comment 11] The abbreviations used in Table 1 should be defined in the table’s legend (BUN, ALT, AST, etc).

Response: Thank you for your suggestion. We have revised the Table legend to define the abbreviations as below:

Lines 218-219:

“WBC: white blood cell, RBC: red blood cell, AST: aspartate aminotransferase, ALT: alanine aminotransferase, BUN: blood urea nitrogen, CRP: C-reactive protein, PCT: procalcitonin.”

[Comment 12] In line 384 of the discussion, it is stated that the findings “might suggest the predominance of increased peripheral platelet consumption”. This statement is very vague. The findings do suggest the predominance of increased peripheral platelet consumption in dengue-induced thrombocytopenia.

Response: Thank you for your suggestion. We have deleted “might” to make our point clear as below.

Lines 406-409:

“… these results suggest the predominance of increased peripheral platelet consumption or destruction over decreased platelet production by bone marrow as an underlying mechanism of dengue-induced thrombocytopenia in the critical and recovery phases.”

[Comment 13] The conclusion drawn in line 389-392 regarding the rate of platelet production by the bone marrow must be couched with some caution, as immature platelets are also susceptible to consumption or destruction, which may lower their absolute count if the rate of platelet loss exceeds the pace of platelet production.

Response: Thank you, we agree and have incorporated your comments in the text as below.

Lines 377-379:

“Moreover, immature platelets are also susceptible to consumption or destruction, which may lower IPF# if the rate of platelet loss exceeds the pace of platelet production.”

Lines 415-416:

“However, the IPF# need to be interpreted with caution because of the susceptivity of immature platelets to consumption or destruction as mentioned above.”

[Comment 14] In line 392, include the IPF% with the IPF# to support the depressed platelet production in CABI with thrombocytopenia.

Response: We agree and have revised the text to include the IPF% with the IPF# as below.

Lines 417-420:

“Because the IPF% and the IPF# of the confirmed CABI group were significantly lower than that of dengue in the critical phase after matching the thrombocytopenia levels, the suppression of platelet production by bone marrow seems to be more predominant in CABI-induced than in dengue-induced thrombocytopenia.”

[Comment 15] In line 401-402, it is stated that the parameters (IPF% and IPF#) are potentially valuable for the differential diagnosis of dengue vs CABI. This should be clarified to be in the differential diagnosis of dengue vs CABI in patients with thrombocytopenia (as the IPF data in non-severe thrombocytopenia patients looks fairly similar to the CABI patients (Fig. 3 vs Fig. 2A)? Further analysis in this regard may be of value to further assess the potential for the IPF to discriminate dengue from CABI/suspected CABI in patients without severe thrombocytopenia.

Response: We agree with your suggestion that, “This should be clarified to be in the differential diagnosis of dengue vs CABI in patients with thrombocytopenia.” We further assessed the potential of the IPF in patients without severe thrombocytopenia (i.e. the subgroups with non-severe thrombocytopenia). The results are shown in a new figure, S4 Fig, entitled “Comparison of platelet parameters among dengue, confirmed CABI, and suspected CABI groups including only the subgroups with non-severe thrombocytopenia by specific time-phases”. The previous S4 Fig has been moved to S5 Fig. According to the new S4 Fig, significant differences of IPF% and IPF# were also shown between dengue and the CABI groups, even in the subgroups with non-severe thrombocytopenia.

This study showed the differences of IPF% and IPF# between dengue and the CABI groups. However, further study is warranted to analyze the performance of IPF (such as sensitivity, specificity, and ROC curve) as a biomarker for the differential diagnosis of dengue vs CABI. We believe that it is not appropriate to further analyze the performance of IPF in this study because there is a possibility that the selection bias could be introduced into the confirmed CABI group as described above (please see our response to the comment 4 of reviewer #1). 

We have included the new S4 Fig with the caption and the texts to further illustrate the comparison including only the subgroups with non-severe thrombocytopenia as below. Furthermore, we have added text to clarify that the potential of IPF as a biomarker for the differential diagnosis was shown especially in patients with thrombocytopenia and further study is needed to analyze the performance of IPF.

Lines 297-300:

“S4 Fig shows a comparison including only the subgroups with non-severe thrombocytopenia and significant differences of IPF% and IPF# were also shown between dengue and the CABI groups in similar phases.”

Lines 424-427:

“IPF% and IPF# are potentially valuable parameters for the differential diagnosis of dengue from CABI especially in patients with thrombocytopenia. Further study is warranted to analyze the performance of IPF as a biomarker for the differential diagnosis of dengue from CABI.”

Lines 593-605:

“S4 Fig. Comparison of platelet parameters among dengue, confirmed CABI, and suspected CABI groups including only the subgroups with non-severe thrombocytopenia by specific time-phases. 

Box and whisker plots show the platelet parameters of dengue, confirmed CABI, and suspected CABI groups observed in specific time-phases: febrile phase (day 1 to 3), critical phase (day 4 to 6), and recovery phase (day 7 to 10). Non-severe thrombocytopenia was defined as a platelet nadir ≥50×103/µl during hospitalization. Boxes show the median and interquartile values, whiskers represent the upper and lower adjacent values and dots indicate outside values. Comparison of dengue, confirmed CABI, and suspected CABI groups were performed using the Kruskal-Wallis test followed by the Dunn’s post hoc test with Holm adjustment. *: P<0.05, †: P<0.01, ‡: P<0.001. CABI: community acquired bacterial infection, IPF%: Immature Platelet Fraction, IPF#: Immature Platelet Fraction Count.”

[Comment 16] The statement from line 402-403 that the parameters (IPF% and IPF#) could be used to predict severity of dengue-induced thrombocytopenia and anticipation of platelet recovery seems unsupported by the data.

Response: We agree with your suggestion and have deleted the part regarding severity prediction and platelet recovery anticipation. The new text is as below.

Lines 423-426:

“Our results show differences of IPF% and IPF# between dengue and the CABI groups. IPF% and IPF# are potentially valuable parameters for the differential diagnosis of dengue from CABI especially in patients with thrombocytopenia.”

Reviewer #2

“The study provides information that is locally relevant on the applicability of novel parameters that would be routinely available and adds to the understanding of the pathophysiology of the disease process. This information could possible serve to risk stratify patients in resource poor environments.”

Thank you very much for your thoughtful feedback. We have revised the points you kindly raised in the PDF file labeled “PONE-D-21-17692 04_07_21 SL.pdf”. We hope that our edits satisfactorily address all the issues and concerns you have noted.

Comments:

[Comment 1] (Line 107 of the previous text) “…on the grounds of …???”

Response: Because many active tuberculosis cases were diagnosed clinically at the study site because of the limited resources, we believe that the original sentence, “Subjects were excluded if suspected of active tuberculosis”, reflects the actual situation at the study site.

Line 121 (these line numbers correspond to the line numbers in the manuscript without tracked changes):

“Subjects were excluded if suspected of active tuberculosis …”

[Comment 2] (Lines 116-117 of the previous text) “Change [Patients were also diagnosed with dengue if positive for dengue IgM without other suspected diagnoses or positive blood culture results.] to [and if result for dengue IgM on serological assay was positive.]”

Response: To increase specificity, we confirmed that there was no laboratory positive results of other diseases or positive blood culture results when we got positive results of dengue IgM. We have rewritten the texts to clarify this point, as below:

Lines 129-131:

“Patients were also diagnosed with dengue if positive for dengue IgM without laboratory positive results of other diseases or positive blood culture results.”

[Comment 3] (Lines 117-129 of the previous text) “(Regarding the diagnostic criteria of CABIs) Why are these other infections of relevance?”

Response: We believe that these diagnostic criteria are necessary because we defined each community acquired bacterial infection (CABI) according to these diagnostic criteria. However, to improve readability, we have moved these diagnostic criteria to S1 appendix in the supporting information.

Lines 127-128:

“The confirmed diagnoses of each disease were as follows: Dengue diagnosis was when the nonstructural protein 1 (NS1) antigen was positive and/or …”

Lines 131-134:

“Diagnostic criteria of each CABI are described in S1 appendix. The CABI that fulfilled the diagnostic criteria was categorize as confirmed CABI. Suspected CABI was a clinical diagnosis after excluding dengue infection but unable to reach a confirmed CABI diagnosis described in S1 appendix.”

[Comment 4] (Lines 141-147 of the previous text) “(Regarding the laboratory procedure for diagnoses of CABIs other than dengue) Relevance?”

Response: We believe that the laboratory procedure for diagnoses of CABIs is necessary because of the same reason described above (please see our response to the comment 3 of reviewer #2). However, to improve readability, we have moved the laboratory procedure for diagnoses of CABIs to S2 appendix in the supporting information.

Lines 146-147:

“Laboratory procedure for diagnoses of CABIs are described in S2 appendix.”

The following are the points revised in accordance with the suggestions of reviewer #2 (indicated with underline or strike-through line).

Line 1-3:

“Full title: Unique characteristics of new complete blood count parameters, the Immature Platelet Fraction and the Immature Platelet Fraction Count, in dengue patients.”

Lines 56-58:

“The purpose of this observational study was to examine thrombopoiesis as reflected by these 2 new CBC parameters in patients infected with dengue. The study was conducted in infectious disease referral hospital in Metro Manila, the Philippines.”

Lines 58-60:

“We enrolled hospitalized patients at admission who were diagnosed with acute dengue or community acquired bacterial infection (CABI).”

Lines 62-63:

“The participants consisted of 152 patients with dengue infection, …”

Lines 64-66:

“At admission, the percent IPF (IPF%) of the patients with dengue was significantly higher than that of the confirmed CABI patients (median 3.7% versus 1.9%; p <0.001).”

Lines 66-67:

“In a time course evaluation, there was no significant difference of IPF% between the patients with dengue infection and the confirmed CABI patients …”

Lines 68-70:

“the IPF% of the patients with dengue infection increased to be significantly higher than that of the confirmed CABI patients …”

Lines 71-72:

“Our study elucidated the unique characteristics and time-course trends of IPF percent and number (IPF#) in the patients with dengue infection.”

Lines 72-74:

“IPF% and IPF# are potentially valuable parameters in dengue and further investigation is required for the optimal use in clinical practice.”

Lines 80-81:

“Thrombocytopenia induced by infection with dengue is typical around the time of defervescence …”

Lines 88-91:

“Various studies have evaluated the utility of these parameters in the evaluation of patients with haematological conditions such as idiopathic thrombocytopenic purpura (ITP), thrombotic thromboctopenic purpura (TTP), aplastic anaemia and chemotherapeutic related thrombocytopenia.”

Lines 95-96:

“… the benefit of evaluating these parameters in patients with dengue infection has not been fully determined.”

Lines 96-99:

“Therefore, this study aimed to investigate the thrombopoietic activity in patients with dengue infection by quantifying IPF% and IPF# and to elucidate their characteristics by comparing with these parameters in patients with community acquired bacterial infection (CABI).”

Lines 109-110:

“ leading training hospital …”

Lines 114-115:

“The participants were subsequently categorized into the three groups - confirmed dengue infection, …”

Line 133:

“Suspected CABI was a clinical diagnosis after excluding dengue infection …”

Lines 153:

“Sysmex XN-1000TM”

Lines 161-163:

“Clinical information consisting of Age demographic data and past medical history were documented at admission. by medical doctors and research nurses using a structured report form. The results of the admission and subsequent CBC results of the patients were recorded in an electronic data base.”

Lines 163-165:

“To evaluate the time course trend of platelet parameters, the representative values of each parameter by day of illness were calculated using only the results of participants who underwent blood tests on a specific day of illness.”

Lines 165-167:

“The timing of blood collection varied between participants and the sequential day-by-day test results were not available for everyone.”

Lines 168-169:

“The clinicians or the researchers were not blinded to the laboratory results.”

Lines 171-173:

“Comparison of the results of dengue, confirmed CABI, and suspected CABI patients were performed using the Kruskal-Wallis test followed by the Dunn’s post hoc test with Holm adjustment.”

Lines 198-199:

“The participants consisted of 152 patients suffering with dengue, 180 confirmed CABI and 274 suspected CABI patients.”

Lines 199-207:

“The confirmed diagnosis in the CABI patients included leptospirosis (59 patients), X-ray confirmed pneumonia (37 patients), bacteremia (12 patients), diphtheria (25 patients), meningococcal disease (14 patients), and skin infection (33 patients). The clinical diagnoses in the suspected CABI patients consisted of pneumonia (72 patients), enteric fever (26 patients), urinary tract infection (23 patients), leptospirosis (10 patients), central nervous system infection (8 patients), abdominal infection (4 patients), meningococcal disease (1 patient), septic rash (1 patient), and undiagnosable infectious disease (129 patients).”

Lines 208-209:

“The median age was 19.0 years (interquartile range (IQR): 13.0, 25.0) in the dengue group, …”

Line 213 (in Table 1):

“Co-morbid conditions”

Lines 220-222:

“2. Comparison of platelet, IPF%, and IPF# at admission between the dengue, confirmed CABI, and suspected CABI groups”

Lines 223-224:

“Fig 1 shows the comparison of platelet, IPF% and IPF# at admission between the dengue, confirmed CABI and suspected CABI groups.”

Lines 362-365:

“In brief, in a time course evaluation the IPF% of the patients with dengue infection was significantly higher than those of the confirmed and suspected CABI groups at admission and during the critical and recovery phases.”

Line 421:

“Drawing from their utility in hematologic disorders, …”

Lines 445-449:

“This study demonstrated the distinctive characteristics and time-course trends of IPF% and IPF# in a dengue group, which were significantly different compared with those of confirmed and suspected CABI groups. IPF% and IPF# are potentially valuable parameters in dengue and further investigation is required for the optimal use in clinical practice.”

Line 563:

“BacT/ALERT®”

Line 582-583:

“The mean of each platelet parameter by days of illness were calculated only from the results of participants who underwent a blood test on each day.”

Figures and Table:

To improve the manuscript according to the reviewers’ comments, we have changed the figure composition as described below:

A new figure has been inserted as S4 Fig.

The previous S4 Fig has been moved to S5 Fig.

---

## [Decision Letter · Decision Letter 1]

11 Oct 2021

Unique characteristics of new complete blood count parameters, the Immature Platelet Fraction and the Immature Platelet Fraction Count, in dengue patients.

PONE-D-21-17692R1

Dear Dr. Yasuda,

We’re pleased to inform you that your manuscript has been judged scientifically suitable for publication and will be formally accepted for publication once it meets all outstanding technical requirements.

Kind regards,

Elizabeth S. Mayne, M.D.

Academic Editor

PLOS ONE

Additional Editor Comments (optional):

Reviewers' comments:

Reviewer's Responses to Questions

**Comments to the Author**

1. If the authors have adequately addressed your comments raised in a previous round of review and you feel that this manuscript is now acceptable for publication, you may indicate that here to bypass the “Comments to the Author” section, enter your conflict of interest statement in the “Confidential to Editor” section, and submit your "Accept" recommendation.

Reviewer #1: (No Response)

Reviewer #2: All comments have been addressed

2. Is the manuscript technically sound, and do the data support the conclusions?

Reviewer #1: Yes

Reviewer #2: Yes

3. Has the statistical analysis been performed appropriately and rigorously? 

Reviewer #1: Yes

Reviewer #2: Yes

4. Have the authors made all data underlying the findings in their manuscript fully available?

Reviewer #1: Yes

Reviewer #2: Yes

5. Is the manuscript presented in an intelligible fashion and written in standard English?

Reviewer #1: Yes

Reviewer #2: Yes

6. Review Comments to the Author

Reviewer #1: The authors have made substantial improvements to the manuscript, and I would recommend it for publication. There are however a few minor/grammatical corrections which are required:

1) The 1st two sentences of the Introduction currently read as follows:

“Dengue is a mosquito-borne viral infection and numerically one of the most important viral diseases in tropical area. [1,2] A total of 390 million dengue infections are estimated per year, and 3.97 billion people are estimated to be at risk of dengue infection worldwide.”. These would be better phrased as follows:

“Dengue is a mosquito-borne viral infection and is one of the most important viral diseases in tropical areas. [1,2] A total of 390 million dengue infections are estimated to occur per year, and 3.97 billion people are estimated to be at risk of dengue 80 infection worldwide.”

2) The last sentence of the Introduction currently reads as follows: “CABI can induce thrombocytopenia the same as dengue and it has clinically significant differences from dengue; specifically, the need for antibiotic treatment and poorer prognosis when accompanied by thrombocytopenia.” This would be better phrased as follows:

“CABI-associated thrombocytopenia has clinically significant differences from dengue; specifically, the need for antibiotic treatment and a poorer prognosis.”

3) In line 117, the following: “Because the limited diagnostic techniques were available at the study site…” should be edited as follows: “Because limited diagnostic techniques were available at the study site…”

4) In line 127, the following: “Dengue diagnosis was when the nonstructural protein 1…” should be edited as follows: “Dengue diagnosis was established when the nonstructural protein 1…”

5) Figure 1 has no image for the platelet count in the severe thrombocytopenia group.

6) In lines 362-365, the following is stated: “In brief, in a time course evaluation the IPF% of the patients with dengue infection was significantly higher than those of the confirmed and suspected CABI groups at admission and during the critical and recovery phases.” However, in Figure 3, the IPF% is not significantly different at the febrile stage (i.e. at admission). The same applies in line 366…

Reviewer #2: The comments of the reviewers have been adequately addressed and the current version of the article is publishable and could improve patient outcomes.

7. PLOS authors have the option to publish the peer review history of their article (what does this mean?). If published, this will include your full peer review and any attached files.

Reviewer #1: No

Reviewer #2: No

---

## [Editor Report · Acceptance letter]

18 Oct 2021

PONE-D-21-17692R1 

Unique characteristics of new complete blood count parameters, the Immature Platelet Fraction and the Immature Platelet Fraction Count, in dengue patients. 

Dear Dr. Saito:

I'm pleased to inform you that your manuscript has been deemed suitable for publication in PLOS ONE. Congratulations! Your manuscript is now with our production department. 

Kind regards, 

on behalf of

Dr. Elizabeth S. Mayne 

Academic Editor

PLOS ONE